# FBNetGen: Task-aware GNN-based fMRI Analysis via Functional Brain Network Generation

**Xuan Kan**[1]                                                              XUAN.KAN@EMORY.EDU
**Hejie Cui**[1]                                                               HEJIE.CUI@EMORY.EDU
**Joshua Lukemire**[2]                                          JOSHUA.LUKEMIRE@EMORY.EDU
**Ying Guo**[2]                                                                 YGUO2@EMORY.EDU
**Carl Yang**[1]                                                          J.CARLYANG@EMORY.EDU

[1] *Department of Computer Science, Emory University*

[2] *Department of Biostatistics and Bioinformatics, Emory University*

**Editors:** Under Review for MIDL 2022

## Abstract

Functional magnetic resonance imaging (fMRI) is one of the most common imaging modalities to investigate brain functions. Recent studies in neuroscience stress the great potential of functional brain networks constructed from fMRI data for clinical predictions. Traditional functional brain networks, however, are noisy and unaware of downstream prediction tasks, while also incompatible with the deep graph neural network (GNN) models. In order to fully unleash the power of GNNs in network-based fMRI analysis, we develop FBNET-GEN, a task-aware and interpretable fMRI analysis framework via deep brain network generation. In particular, we formulate (1) prominent region of interest (ROI) features extraction, (2) brain networks generation, and (3) clinical predictions with GNNs, in an end-to-end trainable model under the guidance of particular prediction tasks. Along with the process, the key novel component is the graph generator which learns to transform raw time-series features into task-oriented brain networks. Our learnable graphs also provide unique interpretations by highlighting prediction-related brain regions. Comprehensive experiments on two datasets, i.e., the recently released and currently largest publicly available fMRI dataset Adolescent Brain Cognitive Development (ABCD), and the widely-used fMRI dataset PNC, prove the superior effectiveness and interpretability of FBNETGEN. The implementation is available at https://github.com/Wayfear/FBNETGEN.

**Keywords:** fMRI, Brain Network, Graph Generation, Graph Neural Network

## 1. Introduction

In recent years, network-oriented analysis has become increasingly important in neuroimaging studies in order to understand human brain organizations in healthy as well as diseased individuals (Satterthwaite et al., 2015; Wang et al., 2016; Wang and Guo, 2019; Bullmore and Sporns, 2009; Deco et al., 2011). There are abundant findings in neuroscience research showing that neural circuits are key to understanding the differences in brain functioning between populations, and the disruptions in neural circuits largely cause and define brain disorders (Insel and Cuthbert, 2015; Williams, 2016). Functional magnetic resonance imaging (fMRI) is one of the most commonly used imaging modalities to investigate brain function and organization (Ganis and Kosslyn, 2002; Lindquist, 2008; Smith, 2012). There is a strong interest in the neuroimaging community to predict clinical outcomes or classify

individuals based on brain networks derived from fMRI images (Kawahara et al., 2017; Yahata et al., 2016). Among them, gender prediction is a fundamental yet very important task for understanding human brains. Although extensive research has revealed sex differences in human cognition, the neural origin of those differences is still obscure and explicit interpretations are in prompt need to advance the biomedical discoveries in the neuroscience field (Satterthwaite et al., 2015; Gur and Gur, 2016).

Current brain network analyses are typically composed of two steps (Smith et al., 2011; Simpson et al., 2013; Wang et al., 2016). The first step is functional brain networks generation from individuals' fMRI data. This is usually done by selecting a brain atlas with a set of regions of interest (ROI) as nodes and extracting fMRI blood-oxygen-level-dependent (BOLD) signal series from each brain region. For the edge generation, pairwise connectivity is calculated between node pairs using measures such as Pearson correlation and partial correlation. Then in the second stage, the obtained brain connectivity measures between all the node pairs are used to classify individuals or predict clinical outcomes.

Although the correlation score-based brain network generation mechanism is widely used in existing literature, it brings two flaws. First, correlation methods focus on capturing linear correlation and ignore temporal order, which means shuffling the time steps does not change the results. Second, there is a growing trend in applying graph neural networks (GNNs) on brain connectivity matrices from fMRI (Yan et al., 2019; Anirudh and Thiagarajan, 2019; Li et al., 2019, 2021), whereas the mechanism of most GNNs (*i.e.*, message passing) is incompatible with existing functional brain networks which possess both positive and negative weighted edges. Hence, these works have to design additional processes in GNNs to handle negative weights. As closer to us, Zhang et al. (Zhang and Huang, 2019a) apply GNNs on brain networks without pre-computed region-to-region correlations, but their randomly generated graphs do not provide biological insights into the structure of brain networks. More detailed related work can be found in Appendix A.

In this work, to unleash the power of GNNs in network-based fMRI analysis while providing valuable interpretability regarding brain region connectivity, we propose to generate functional brain networks from fMRI data that are compatible with GNNs and customized towards downstream clinical predictions. Specifically, we develop an end-to-end differentiable pipeline from BOLD signal series to clinical predictions. Our pipeline includes three components: a time-series encoder for dimension reduction and denoise of raw time-series data, a graph generator for individual brain networks generation from the encoded features, and a GNN predictor for clinical predictions based on the generated brain networks (c.f. Figure 1). Compared with existing methods, our proposed approach demonstrates three benefits: (a) We adopt commonly used deep sequence models as the encoder for time-series which can capture both non-linear and temporal relations. (b) Our generated graphs only contain positive values which are compatible with most existing GNNs. Besides, regularizers for the generator are designed to maintain the consistency within a class and maximize the difference among classes. (c) Our generated graphs are dynamically optimized towards the downstream tasks, thus providing explicit interpretations for biomedical discoveries.

We conduct extensive experiments using real-world fMRI datasets with the important downstream task of gender prediction. FBNetGen achieves consistently better accuracy over four types of baselines. Furthermore, our in-depth analysis identifies a set of brain regions useful for gender prediction, aligning well with existing neurobiological findings.

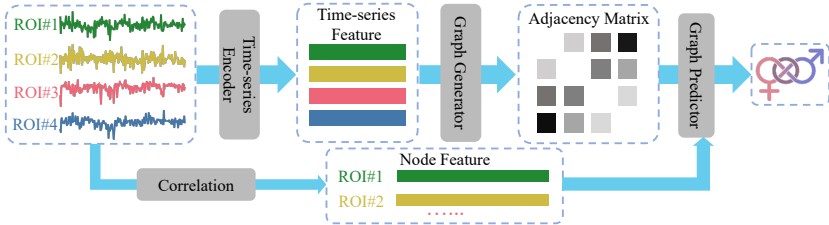

Figure 1: The overall framework of our proposed end-to-end task-aware fMRI analysis via functional brain network generation.

## 2. FBNetGen

### 2.1. Overview

In this section, we elaborate on the design of FBNetGen and its three main components, as is shown in Figure 1. Specifically, the input $\boldsymbol{X} \in \mathbb{R}^{n \times v \times t}$ denotes the BOLD time-series for regions of interest (ROIs), where $n$ is the sample size, $v$ is the number of ROIs and $t$ is the length of time-series. Each $\boldsymbol{x} \in \mathbb{R}^{v \times t}$ represents the time series of a sample. The target output is the prediction label $\boldsymbol{Y} \in \mathbb{R}^{n \times |\mathcal{C}|}$, where $\mathcal{C}$ is the class set of $Y$ and $|\mathcal{C}|$ is the number of classes. Besides, we also generate a functional brain network $\boldsymbol{A} \in \mathbb{R}^{v \times v}$ (i.e., brain connectivity matrix) for each sample $\boldsymbol{x} \in \mathbb{R}^{v \times t}$ as an intermediate of the end-to-end prediction pipeline. The generated graph highlights prediction-specific prominent brain network connections and provides unique interpretation towards neuroscience research.

### 2.2. Time-series Encoder

The input BOLD signal series is a type of temporal sequence data. The main difference between BOLD signal series with ordinary time-series data is that BOLD is a group of aligned sequences instead of independent ones. Traditional BOLD encoder methods like ICA (Tuovinen et al., 2017) and PCA (Thomas et al., 2002) ignore the temporal order. Also, PCA and ICA can only capture linear information within or across time series. Besides, to construct brain networks from ICA and PCA results, Pearson correlation is often adopted (Smith et al., 2011). However, brain networks generated in this way are not aware of the downstream tasks and not compatible with GNNs, due to the negative edge weights.

Recently, deep neural networks have shown great success in capturing complex non-linear information on various time-sequence tasks, such as natural language process, market analysis and traffic control (Shang et al., 2021; Chung et al., 2014; Kiranyaz et al., 2019). In this work, we focus on the novel pipeline of generating predictive and interpretable brain networks, so we simply adopt two commonly used deep encoding models, 1D-CNN and bi-GRU. Specifically, we choose bi-GRU rather than the well-known LSTM since the former one can achieve similar performance with fewer parameters (Chung et al., 2014). In brain network analyses where the sample size is usually pretty small (e.g., less than 1000), such lightweight frameworks can be essential to avoid over-fitting. The overall process of extracting the region feature from time-series data for each ROI can be formulated as $\boldsymbol{h}_e = \text{Encoder}(\boldsymbol{x}), \boldsymbol{h}_e \in \mathbb{R}^{v \times d}$. For more details about the design of 1D-CNN and bi-GRU

encoder, please refer to Appendix C. Based on our empirical results, even such simple encoders can clearly attribute the superiority of our generated graphs.

## 2.3. Graph Generator

Based on the encoded time-series feature $\boldsymbol{h}_e$, a task-oriented functional brain network is generated for the subsequent predictor. It is formulated as the connectivity matrix $\boldsymbol{A}$, which stores the pairwise connectivity strengths between ROIs as its elements. Unlike the commonly used traditional approach for functional brain network construction that calculates the pairwise Pearson correlations between raw time-series of ROIs (Smith et al., 2011), we generate a learnable graph as $\boldsymbol{A} = \boldsymbol{h}_A \boldsymbol{h}_A^T$, where $\boldsymbol{h}_A = \text{softmax}(\boldsymbol{h}_e)$. The learnable graph $\boldsymbol{A}$ can be regularized by the downstream prediction task in an end-to-end framework. The softmax operation highlights the strong ROI connections by generating skewed positive edge weights, which are compatible with GNNs and valuable for interpretation. In contrast, brain connectivity matrices generated from traditional (i.e., statistical) methods are not compatible with GCN, since they contain negative weights in edges.

Considering the limited supervision in neuroscience research, sheer supervision is usually not enough to fit a model very well. For instance, the number of parameters in the used 1D-CNN encoder can reach up to 20k, while there are only 353 samples in PNC training set. It becomes even harder for the model to generate high-quality graphs when the gradient feedback in the end-to-end framework is too long.

In order to facilitate the learning of brain networks beyond the sheer supervision of graph-based classification, we consider incorporating exterior regularizations that are in line with scientific studies. Existing literature has concluded that (Tunç et al., 2016; Satterthwaite et al., 2015) edge-level difference exists between genders in both structural and functional brain connectivity matrices. Inspired by this observation, we further design three regularizers and apply them to the generator model during training, named group intra loss, group inter loss and sparsity loss, respectively.

**Group intra loss.** The set of samples with the same prediction label is called a group. Previous clinical findings (Sporns, 2013) show that there are consistent patterns in resting-state functional connectivity among individuals from the same group. In order to utilize the latent consistent patterns as a regularization, we propose a group intra loss, which aims to minimize the difference among the connectivity matrices within a class. Given a class $c \in \mathcal{C}$ and the set $\mathcal{S}^c = \{i \mid Y_{i,c} = 1\}$ containing all the indexes of samples with label $c$, the mean $\mu_c$ and variance $\sigma_c^2$ of their learnable graphs $\boldsymbol{A}$ within a batch are calculated as

$$\mu_c = \sum_{k \in \mathcal{S}^c} \frac{\boldsymbol{A}^k}{|\mathcal{S}^c|}, \ \sigma_c^2 = \sum_{k \in \mathcal{S}^c} \frac{\left\| \boldsymbol{A}^k - \mu_c \right\|_2^2}{|\mathcal{S}^c|}. \tag{1}$$

Based on some derivations, the group intra loss can be effectively calculated in $\mathcal{O}(n)$ as

$$L_{intra} = \sum_{c \in \mathcal{C}} \sum_{i \in \mathcal{S}^c} \frac{\left\| \boldsymbol{A}^i - \mu_c \right\|_2^2}{|\mathcal{S}^c|} = \sum_{c \in \mathcal{C}} \sigma_c^2. \tag{2}$$

**Group inter loss.** Cognition science findings in (Satterthwaite et al., 2015) substantiate that there are significant differences among the functional brain networks across different

genders, such as the brain volume. Hence, we incorporate a group inter loss that aims to maximize the difference of connectivity matrices across different groups, while keeping those within the same group similar. This inter loss can also be calculated in $\mathcal{O}(n)$ time as

$$L_{inter} = \sum_{a,b \in \mathcal{C}} (\sigma_a^2 + \sigma_b^2 - \frac{\sum_{i \in \mathcal{S}^a} \sum_{j \in \mathcal{S}^b} \left\| \boldsymbol{A}^i - \boldsymbol{A}^j \right\|_2^2}{|\mathcal{S}^a||\mathcal{S}^b|}) = -\sum_{a,b \in \mathcal{C}} \left\| \mu_a - \mu_b \right\|_2^2. \tag{3}$$

**Sparsity loss.** The model with only group losses may overemphasize the graph difference between genders, which could harm the model's performance and stabilization. To mitigate the degree of deviation caused by large values in the generated graph and highlight the most contributory task-specific ROI connections, we further enforce the sparsity of generated brain networks, where a sparsity loss is formulated as

$$L_{sparsity} = \frac{1}{vv} \sum_{i=1}^{v} \sum_{j=1}^{v} \boldsymbol{A}^{ij}. \tag{4}$$

### 2.4. Graph Predictor

For the graph predictor, we adopt GNNs, which learn node representations by transforming and propagating node features and structure information on the constructed graphs.

In practice, we initialized node feature $\boldsymbol{F}_p$ for the node $p$ as a vector of Pearson correlation scores between its time-series with all the nodes contained in the graph. With the initial node features $\boldsymbol{F} \in \mathbb{R}^{v \times f}$ of ROIs and the learnable connectivity matrix $\boldsymbol{A}$ from the graph generator, we can apply a $k$-layer graph convolutional network (Kipf and Welling, 2017) to update the node embeddings through $\boldsymbol{h}^k = \text{ReLU}\left(\boldsymbol{A}\boldsymbol{h}^{k-1}W^k\right)$, where $W^k$ represents learnable parameters in convolutional layers and $\boldsymbol{h}^0 = \boldsymbol{F}$. Since the order of nodes is fixed, the permutation invariance is not a concern on the brain graphs. The graph-level embedding can be obtained by concatenating all the node embeddings after the final convolutional layer. The influence of performance with different pooling strategies can be found in Appendix E. A BatchNorm1D is further applied to avoid extremely large values. Finally, another MLP layer is employed for class prediction, $\hat{y} = \text{MLP}\left(\text{BatchNorm1D}\left(\|_{p=1}^v \boldsymbol{h}_p^k\right)\right)$.

### 2.5. End-to-end Training

We combine the aforementioned three components into an end-to-end framework, where the label $y$ and the task-oriented graphs are leveraged at the same time. Another advantage of end-to-end training is that the feature encoder provides a larger parameter search space than a pure GNN prediction model, leading to potential performance improvements. Overall, our final training objective is composed of four terms: $L = L_{ce} + \alpha L_{intra} + \beta L_{inter} + \gamma L_{sparsity}$, where $L_{ce}$ is the supervised cross-entropy loss for prediction, and $\alpha, \beta, \gamma$ are tunable hyperparameters representing the weights of three regularizers respectively.

### 3. Experiments

In this section, we evaluate the effectiveness and interpretability of FBNetGen with extensive experiments. For effectiveness, we compare FBNetGen with four types of baselines

and demonstrate the effectiveness of each loss term with the ablation study. For interpretability, we aim to investigate the advantages of FBNetGen regarding the consistency between important brain connectivity patterns in our learned graphs and the existing neuroscience discoveries. The influence of hyper-parameters is attached in Appendix D.

### 3.1. Experimental Settings

**Dataset.** We conduct experiments on two real-world fMRI datasets. (a) *Philadelphia Neuroimaging Cohort (PNC)* includes individuals aged 8–21 years (Satterthwaite et al., 2014). We utilize rs-fMRI data of 503 subjects, with 289 (57.46%) of them being females. The region definition is based on 264-node atlas (Power et al., 2011). (b) *Adolescent Brain Cognitive Development Study (ABCD)* (Casey et al., 2018) is one of the largest publicly available fMRI datasets. We include 7901 children and 3961 (50.1%) among them are female. The region definition is based on the HCP 360 ROI atlas (Glasser et al., 2013).

**Metrics.** We choose gender prediction as the evaluation task, with available labels from both PNC and ABCD datasets. Considering gender prediction is a binary classification problem and both datasets are balanced across classes, AUROC is the most comprehensive performance metric and is adopted here for a fair comparison. Besides, we also include accuracy as a metric to reflect the practical prediction performance of FBNetGen. All the reported performances are the average result of 5 runs on the test set.

Please refer to Appendix B for details about datasets, metrics and implementation.

### 3.2. Performance Comparison and Ablation Studies

We compare FBNetGen with baselines of four types. To ensure fairness, grid search is applied to all baselines and the best one is reported for comparison.

**(a) Models directly using time-series features.** We compare our graph-based model with two baselines modeling time-series data without graph construction. Two commonly used models for temporal sequence, 1D-CNN and bi-GRU, are applied to encode BOLD time-series, which share the same architectures as the feature encoder of FBNetGen. **(b) Models using traditional methods to construct graphs.** To investigate the advantage of our task-oriented learnable graph generator over traditional statistics-based methods, we compare our task-oriented graphs $\boldsymbol{A}$ with two widely practiced ways to construct brain networks. To isolate the influence of node features, we also use the uniform graphs $\boldsymbol{A}^U$ as the control group, which sets all the entries in the adjacency matrixes with one. All these two types of graphs are paired with the same node features $\boldsymbol{F}$ as FBNetGen and then processed by the same GNN predictor for downstream prediction. **(c) Models using other learnable graph generators.** We further introduce three other baselines based on learnable graph generators, namely LDS-GRU, LDS-CNN and GTS. LDS (Franceschi et al., 2019) is a framework which jointly learns graph structures and model parameters through bilevel optimization when graph structures are not available. GTS is another existing method that can learn graph structures from a group of time-series data (Shang et al., 2021). It combines time-series data and graph structures to make sequence-to-sequence predictions. Here GTS is revised to generate classification results for each sample to suit our gender prediction task. **(d) Other deep learning models for brain networks.** We also compare our method with two popular deep models for brain

networks, BrainnetCNN(Kawahara et al., 2017) and BrainGNN(Li et al., 2021). Besides, FCNet(Riaz et al., 2017), a method that uses deep learning to generate functional brain networks, is compared as a baseline.

Table 1: Performance comparison with three types of baselines

| Type | Method | Dataset: PNC | | Dataset: ABCD | |
|---|---|---|---|---|---|
| | | AUROC | Accuracy | AUROC | Accuracy |
| Time-series | 1D-CNN | $63.7 \pm 3.8$ | $54.7 \pm 1.2$ | $68.1 \pm 3.1$ | $63.2 \pm 2.6$ |
| | bi-GRU | $65.1 \pm 3.5$ | $58.1 \pm 2.4$ | $51.2 \pm 1.0$ | $49.9 \pm 0.8$ |
| Traditional Graph | GNN-Uniform | $70.6 \pm 4.8$ | $66.2 \pm 3.9$ | $88.8 \pm 0.7$ | $80.5 \pm 0.7$ |
| | GNN-Pearson | $69.6 \pm 4.5$ | $65.6 \pm 3.3$ | $88.8 \pm 0.3$ | $80.7 \pm 0.6$ |
| Learnable Graph | LDS-GRU | $76.9 \pm 3.2$ | $72.2 \pm 5.8$ | $90.1 \pm 0.5$ | $81.4 \pm 1.6$ |
| | LDS-CNN | $78.2 \pm 3.8$ | $70.8 \pm 6.2$ | $90.7 \pm 0.3$ | $82.5 \pm 0.9$ |
| | GTS | $68.2 \pm 1.9$ | $63.7 \pm 2.4$ | $87.8 \pm 1.1$ | $76.7 \pm 2.0$ |
| Deep Model | BrainnetCNN | $78.5\pm3.2$ | $71.9\pm4.9$ | $93.5\pm0.3$ | $85.7\pm0.8$ |
| | BrainGNN | $77.5\pm3.2$ | $70.6\pm4.8$ | OOM | OOM |
| | FCNet | $53.4\pm3.7$ | $55.7\pm2.4$ | $50.2\pm0.4$ | $51.3\pm0.6$ |
| Ours | FBNETGNN-CNN | $80.7\pm4.7$ | $74.0\pm6.0$ | $93.9\pm0.8$ | $86.3\pm0.7$ |
| | FBNETGNN-GRU | $\mathbf{80.8\pm3.3}$ | $\mathbf{74.8\pm2.4}$ | $\mathbf{94.5\pm0.7}$ | $\mathbf{87.2\pm1.2}$ |

It can be seen from Table 1 that our FBNETGEN outperforms both time-series and traditional graph baselines with the same encoder by large margins. Furthermore, FBNET-GEN consistently surpasses learnable graph baselines and other deep models, demonstrating the superior advantage of our proposed task-aware brain network generation.

Table 2: AUROC performance with different regularizers

| Dataset | PNC | | | | ABCD | | | |
|---|---|---|---|---|---|---|---|---|
| Regularizers | All | CE | CE+GL | CE+SL | All | CE | CE+GL | CE+SL |
| FBNETGNN-CNN | $\mathbf{80.7\pm4.7}$ | $75.4 \pm 2.7$ | $79.7\pm3.1$ | $80.0\pm3.2$ | $\mathbf{93.9\pm0.8}$ | $91.0 \pm 0.8$ | $93.6\pm0.2$ | $93.2\pm0.4$ |
| FBNETGNN-GRU | $\mathbf{80.8\pm3.3}$ | $75.6 \pm 1.4$ | $80.2\pm3.2$ | $79.4\pm3.3$ | $\mathbf{94.5\pm0.7}$ | $91.3 \pm 0.5$ | $91.8\pm0.4$ | $92.9\pm0.7$ |

We further examine the designed components in our graph generator (Section 2.3): the Group Loss (GL), including both group inner loss and group intra loss, and the Sparsity Loss (SL). We vary the original model with all loss terms, specifically Cross Entropy Loss (CE), GL and SL by removing each component once at a time, and observe the performance of each ablated model variant. The results are shown in Table 2, and training curves are included in Appendix F Figure 4. From the training curves and the final performance, we see that on both PNC and ABCD datasets, the original model with all designed components improves more stably than its three ablated versions and finally achieves the highest performance. Specifically, CE+SL and CE+GL achieves close-to-optimal performance, demonstrating the effectiveness of exterior regularizations in generating informative brain networks.

### 3.3. Interpretability Analysis

In this section, we visualize and compare our generated learnable graphs with the most commonly used existing functional brain networks based on Pearson correlation (Smith

et al., 2011). Our results indicate that our learnable graph approach is more task-oriented and advantageous in capturing differences among classes. Specifically, we use the average

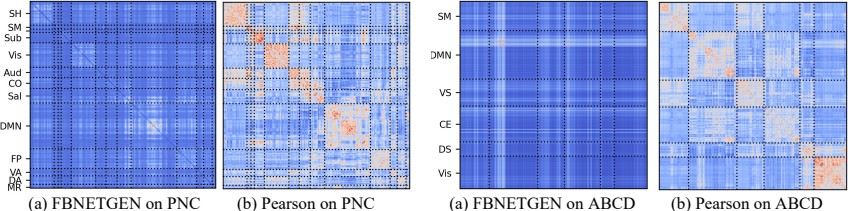

(a) FBNETGEN on PNC     (b) Pearson on PNC     (a) FBNETGEN on ABCD     (b) Pearson on ABCD

Figure 2: Visualizations of our learnable graph vs. Pearson graph. Warmer colors indicate higher values. The full name of each abbreviation can be found in Appendix G.

graph on all samples to demonstrate the predominant neural systems across subjects. The mean heatmap visualization of our learnable brain graphs and the Pearson brain graphs are shown in Figure 2. As is indicated by the heat values, our generated graphs distinctively and consistently highlight the default mode network (DMN) for both the PNC and ABCD datasets. This aligns well with previous neurobiological findings on the PNC data (Satterthwaite et al., 2015), which identify that regions with significant differences between genders are within the DMN. This consistency remains across both datasets and brain atlases, validating that our method yields reliable results reproducible across studies. In contrast, the most significant positive components in the Pearson graphs are the connections within functional modules. These within-module connections reflect intrinsic brain functional organizations but are not necessarily informative for gender prediction.

To demonstrate that our learnable graphs possess better discrimination ability among classes, we divide these learnable graphs based on genders, and apply t-tests to identify edges $\mathcal{E}^d$ with significantly different strengths ($p < 0.05$) across genders. A difference score $T$ is designed to reflect the discrimination ability. Higher scores indicate larger differences between genders. The definition of the score $T$ can be found in Appendix H.1. For the PNC data, the top 3 functional modules are memory retrieval network, default mode network and ventral attention network. For the ABCD data, the top 2 modules are default mode network and ventral salience network (including ventral attention network). Literature (Satterthwaite et al., 2015) indicates that ROIs with significant sex differences are located within the default mode network, the ventral attention network, and the memory retrieval network, which aligns well with our top-ranked modules; whereas Pearson graphs cannot match the literature as well as our graphs. For more details, please refer to Appendix H.2.

## 4. Conclusion

In this paper, we present FBNETGEN, a task-aware GNN-based framework for fMRI analysis via functional brain network generation, which generates the brain connectivity matrices and predicts clinical outcomes simultaneously from fMRI BOLD signal series. Extensive experiments demonstrate that FBNETGEN consistently outperforms four types of existing baselines and provide aligned interpretation results with neurobiological findings.

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

## Appendix A. Background and Related Work

### A.1. fMRI-based Brain Network Analysis

fMRI has become the most commonly used imaging modality to probe brain functional organizations by identifying brain functional networks that represent a set of spatially disjoint regions in the brain, demonstrating coherent temporal dynamics in fMRI blood oxygen level dependent (BOLD) signals (Friston et al., 1993). Functional connectivity (FC) has been found to be related to intrinsic neural processing, cognitive, emotional, visual, and motor functions. Existing studies have shown that FC plays an important role in understanding neurodevelopment, mental disorders and neurodegenerative diseases (Fornito et al., 2015; Ernst et al., 2015; Liu et al., 2008; Craddock et al., 2009; Stam et al., 2007). There are also findings that reveal gender differences in FC between brain regions (Satterthwaite et al., 2015). To investigate FC alterations in demographic and clinical subpopulations, the commonly adopted methods include edge-wise tests for between-group differences (Nichols and Holmes, 2002; Chen et al., 2015; Kim et al., 2015), tests for detecting coordinated disruptions across multiple brain subsystems (Zalesky et al., 2010; Higgins et al., 2019), graph theory based methods for comparing brain network graph metrics (Rudie et al., 2013; Rubinov and Sporns, 2010; Fornito et al., 2013) and graphical model approaches (Lukemire et al., 2021; Higgins et al., 2018; Kundu et al., 2018).

Some recent works have used neural networks to generate FC. For example, FCNet(Riaz et al., 2017) uses neural networks to generate FCMs. However, it is not an end-to-end model. In contrast, it uses ElesticNet to select features and apply an SVM to predict the final outcomes, so its framework requires much engineering and the brain networks they construct are not specific to the prediction task.

### A.2. Graph Neural Networks

Graph Neural Networks (GNNs) have revolutionized the field for modeling various important real-world data in the form of graphs or networks (Kipf and Welling, 2017; Hamilton et al., 2017; Velickovic et al., 2018; Chen et al., 2018; Shi et al., 2021), such as social network, knowledge graphs, protein-interaction networks, etc.The advantage of GNNs is that they can combine node features and graph structures in an end-to-end fashion towards downstream prediction tasks. Various delicately designed GNN models have been developed for graph classification. For example, GCN (Kipf and Welling, 2017) is one of the basic and most representative GNNs that generalized the shared filter for graphs from the successful CNNs models in computer vision; Velickovic et al. (Velickovic et al., 2018) integrated the attention mechanism to assign different weights for neighbors in each graph convolution layers; Xu et al. (Xu et al., 2019) proposed graph isomorphism network, which is a simple yet powerful architecture that is proved to have the equal discriminative ability with the 1-WL test.

Yet, until recently, some emerging attention has been devoted to the generalization of GNN-based models to fMRI-based brain network analysis (Li et al., 2019, 2021). However, GNNs require explicitly given graph structures and node features, which are typically not available in brain networks and are usually constructed manually based on statistical correlations (Smith et al., 2011). In addition, only one recent study has considered the learnable

generation of brain networks but without downstream tasks (Zhang and Huang, 2019b), and no study has explored the interpretability of the generated brain networks towards downstream tasks, which is critical in neuroimaging research regarding its practical scope and promising social impact.

## Appendix B. Experimental Settings

### B.1. Dataset

We conduct experiments demonstrating the utility of FBNETGEN using two real-world fMRI datasets.

The first dataset is from the Philadelphia Neuroimaging Cohort (PNC), a collaborative project from the Brain Behavior Laboratory at the University of Pennsylvania and the Children's Hospital of Philadelphia. It includes a population-based sample of individuals aged 8–21 years (Satterthwaite et al., 2014). After excluding subjects with excessive motion (Satterthwaite et al., 2015; Wang et al., 2016), 503 subjects' rs-fMRI data were included in our analysis. Among these subjects, 289 (57.46%) are female, which indicates that our dataset is balanced across genders. In our paper, we adapt the 264-node atlas defined by (Power et al., 2011) for connectivity analysis. The nodes are grouped into 10 functional modules that correspond to major resting-state networks (Smith et al., 2009). Standard pre-processing procedures are applied to the rs-fMRI data. For rs-fMRI, the pre-processing includes despiking, slice timing correction, motion correction, registration to MNI 2mm standard space, normalization to percent signal change, removal of linear trend, regressing out CSF, WM, and 6 movement parameters, bandpass filtering (0.009–0.08), and spatial smoothing with a 6mm FWHM Gaussian kernel. In the resulting data, each sample contains 264 nodes with time-series data collected through 120 time steps. For connectivity analysis, we focus on the 232 nodes in the Power's atlas that are associated with major resting-state functional modules (Smith et al., 2009).

The second dataset from Adolescent Brain Cognitive Development Study (ABCD) (Casey et al., 2018), is one of the largest publicly available fMRI datasets. This study is recruiting children aged 9–10 years across 21 sites in the U.S. Each child is followed into early adulthood, with repeated imaging scans as well as extensive psychological and cognitive testing. This study started in 2016 and releases data in regular intervals. We use rs-fMRI scans for the baseline visit processed with the standard and open-source ABCD-HCP BIDS fMRI Pipeline [1]. The HCP 360 ROI atlas template is used for each subject's data (Glasser et al., 2013). After processing, each sample contains a connectivity matrix whose size is $360 \times 360$ and BOLD time-series for each node. Since different sample's BOLD time-series have different lengths, only samples with at least 512 time points are selected, and only the first 512 time points for each sample are included in the subsequent analysis. After this selection, 7901 children were included in the analysis. Among them, 3961 (50.1%) are female and 3940 (49.9%) are male. For interpretability analysis, we organize nodes into communities using the AAc-6 parcellation scheme provided by (Akiki and Abdallah, 2019), which divides the 360 ROIs into 6 functional modules.

---

1. https://github.com/DCAN-Labs/abcd-hcp-pipeline

## B.2. Metrics

We choose gender prediction as the evaluation task, with available labels from both PNC and ABCD datasets. Since gender prediction is a binary classification problem and both PNC and ABCD datasets are balanced across classes, AUROC is the most comprehensive performance metric and is adopted here for fair performance comparison. Besides, we also include accuracy as a metric to reflect the practical prediction performance of FBNETGEN.

## B.3. Implementation details

For experiments on the two different feature encoders, we restrict the number of 1D-CNN layer $u$ as 3 since the dataset is relatively small. The detailed design of the 1D-CNN encoder can be found in Table 3. Regarding GRU feature encoder, we set the number of layers to 4. For both feature encoders, the embedding size $d$ of $\boldsymbol{h}_e$ is searched from $\{4, 8, 12\}$, and the window size $\tau$ is tuned from different ranges based on the sequence length of each dataset, with $\{4, 6, 8\}$ for PNC, and $\{8, 16, 32\}$ for ABCD. For the graph generator, the weights of each loss component $\alpha, \beta, \gamma$ are set as $10^{-3}$, $10^{-3}$ and $10^{-4}$, respectively. As for the graph predictor, the number of GCN layers is set as 3 following the common practice. We randomly split 70% of the datasets for training, 10% for validation, and the remained are utilized as the test set. In the training process of FBNETGEN, we use the Adam optimizer with an initial learning rate of $10^{-4}$ and a weight decay of $10^{-4}$. The batch size is set as 16. All the models are trained for 500 epochs and that achieves the highest AUROC performance on the validation set is tested for performance comparison. All the reported performances are the average results of 5 runs. Please refer to the supplementary material for all codes of the implementation of FBNETGEN.

Table 3: 1D-CNN encoder design.

| Layers | Kernal Size | Other Parameters |
|---|---|---|
| Conv 1 | $1 \times \tau \times 32$ | stride=2 |
| Conv 2 | $32 \times 8 \times 32$ | stride=1 |
| Conv 3 | $32 \times 8 \times 16$ | stride=1 |
| Max Pool | 16 | N.A. |
| Flatten | N.A. | N.A. |
| Fully Connected 1 | N.A. | output=32 |
| ReLU | N.A. | N.A. |
| Fully Connected 2 | N.A. | output=8 |

## B.4. Computation complexity

In FBNETGEN, the computation complexity of feature encoder, graph generator and graph predictor are $\mathcal{O}(\mu v t)$, $\mathcal{O}(v^2)$ and $\mathcal{O}(k v^2)$ respectively, where $\mu$ is the layer number of feature encoder, $v$ is the number of ROIs, $t$ is the length of time-series, and $k$ is the layer number of graph predictor. The overall computation complexity of FBNETGEN is thus $\mathcal{O}(v(v + t))$.

## Appendix C.  The Design of Two Encoders

Specifically, when the feature encoder is set as $u$-layer 1D-CNN, the process of generating node features $\boldsymbol{h}_e \in \mathbb{R}^{v \times d}$ for $v$ ROIs can be decomposed as

$$\boldsymbol{h}^u = \text{CONV}_u(\boldsymbol{h}^{u-1}), \ \boldsymbol{h}_e = \text{MLP}(\text{MAXPOOL}(\boldsymbol{h}^u)), \boldsymbol{h}_e \in \mathbb{R}^{v \times d}, \tag{5}$$

where $\boldsymbol{h}^0 = \boldsymbol{x}$ is the original BOLD signal sequence, $d$ is the embedding size of each ROI and the kernal size of $\text{CONV}_1$ equals the window size $\tau$. Similarly, when the feature encoder is set as bi-GRU, the process can be decomposed as

$$\boldsymbol{h}_r = \text{biGRU}([\boldsymbol{x}^{(z\tau-\tau):z\tau}]), \text{where } \boldsymbol{h}_r \in \mathbb{R}^{v \times 2\tau}, z = 1, \cdots, \lfloor \frac{t}{\tau} \rfloor, \tag{6}$$

where $[\boldsymbol{x}^{(z\tau-\tau):z\tau}]$ represents splitting the input sequence $x$ into $z$ segments of length $\tau$. Finally, a MLP layer is applied to generate the final embedding of size $d$ for each ROI

$$\boldsymbol{h}_e = \text{MLP}(\boldsymbol{h}_r), \boldsymbol{h}_e \in \mathbb{R}^{v \times d}. \tag{7}$$

## Appendix D.  Influence of Hyper-parameters

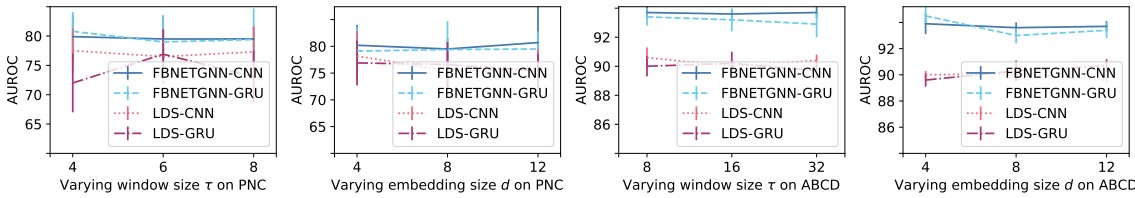

Figure 3:  Influence of two hyper-parameters in the feature encoder of FBNETGEN and LDS based baselines.

We investigate two hyper-parameters that are most influential to the performance of the compared models, namely window size $\tau$ and embedding size $d$ in the feature encoders of 1D-CNN and GRU. To reflect the influence comprehensively, we also include the LDS baselines that also use these feature encoders. The results of adjusting hyper-parameters on PNC and ABCD datasets are shown in Figure 3. As we can observe from Figure 3, increasing the window size and embedding size does not necessarily improve the overall performance of FBNETGEN, demonstrating the stable performance of FBNETGEN. It is impressive that our FBNETGEN consistently achieves better performance compared with the baselines of LDS-CNN and LDS-GRU, in the large ranges of hyper-parameters. Besides, when training data are more sufficient, FBNETGEN outperforms baseline methods with a large margin. This highlights the reliable supremacy of FBNETGEN over other graph generators.

## Appendix E. Influence of Pooling Strategies

To examine the influence pooling function, we vary the original model with different pooling strategies, sum, and concat and observe the performance of each ablated model variant. The results are shown in Table 4. Our results reveal that the concat pooling strategy consistently outperforms the other method across all datasets.

Table 4: AUROC performance with different pooling strategies

| Dataset | PNC | | ABCD | |
|---|---|---|---|---|
| Pooling Strategies | SUM | Concate | SUM | Concate |
| FBNETGNN-GRU | 79.1±4.5 | **80.8±3.3** | 93.6±0.2 | **94.5±0.7** |

## Appendix F. Training Curves of FBNetGen Variants

In Figure 4, we demonstrate the training curves of different FBNETGEN variants. The curves of different variants display similar patterns across datasets. Specifically, it is shown that Group Loss (GL) can achieve pronounced improvement for the model's performance, which proves the effectiveness of our loss design in Section 2.3. Also, applying both exterior regularizers (Group Loss and Sparsity Loss) together with the supervised Cross Entropy loss (CE) can consistently achieve the best performance compared with other settings, representing the importance of the mutually restrictive relationships between different regularizers.

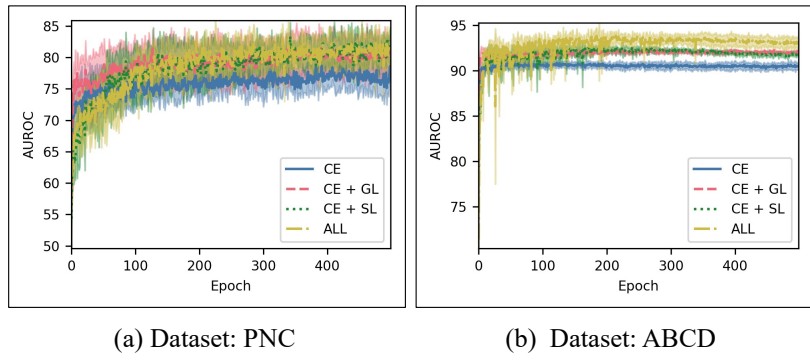

(a) Dataset: PNC      (b) Dataset: ABCD

Figure 4: Training curves of FBNETGEN variants on two datasets.

## Appendix G. Abbreviations of Neural Systems

On PNC, SH (Somatomotor Hand), SM (Somatomotor Mouth), Sub (Subcortical), Vis (Visual), Aud (Auditory), CO (Cingulo-opercular), Sal (Salience), DMN (Default mode), FP (Fronto-parietal), VA (Ventral attention), DA (Dorsal attention), MR (Memory retrieval), On ABCD, SM(Somatomotor), DMN (Default mode), VS (Ventral salience), CE (Central executive), DS (Dorsal salience), Vis (Visual). In ABCD, SM (Somatomotor Mouth), DMN (Default mode), VS (Ventral salience), CE (Central executive), DS (Dorsal salience), Vis (Visual).

## Appendix H. Difference Score $T$ of Functional Modules

### H.1. The Definition of Difference Score $T$

The difference score $T_u$ of each predefined functional module $u$ is calculated as

$$T_u = \sum_{(p,q)\in\mathcal{E}^d} \frac{\mathbb{1}(p \in \mathcal{M}_u) + \mathbb{1}(q \in \mathcal{M}_u)}{2v|\mathcal{M}_u|}, \tag{8}$$

where $v$ is the number of ROIs and $\mathcal{M}_u$ is a set containing all indexes of nodes belonging to the module $u$.

### H.2. Difference Score $T$ of Functional Modules on Learnable Graph and Pearson Graph

The ranked difference score $T$, as defined in Eq. (8), of functional modules on two kinds of graph, learnable and Pearson, on two datasets are shown in Table 5 and Table 6, respectively. Note that the words in bold represent modules that contain more ROIs with significant gender differences according to existing neurobiological findings(Satterthwaite et al., 2015). The ideal case achieves when the modules with higher difference scores are matched with those known important ones from the neuroscience study.

On the PNC dataset, our task-aware learnable graph can obviously highlight the modules with ROIs that are significantly different between genders better compared with Pearson graphs. Regarding the ABCD dataset, due to the fewer functional modules it contains compared with PNC's, the results of two kinds of graphs are similar. However, our learnable graphs put more emphasize on the functional module "Somatomotor", which contains ROIs related to auditory functions that are highly differentiated between genders (Akiki and Abdallah, 2019).

Overall, the difference score comparison of our learnable graphs and Pearson graphs validates that graphs produced by FBNetGen are task-oriented and can capture more authentic differences between genders than existing famous methods.

Table 5: Modules' difference score $T$ of our learnable and Pearson graphs on the PNC dataset.

| Learnable Graph | | Pearson Graph | |
|---|---|---|---|
| Module | Difference Score | Module | Difference Score |
| **Memory retrieval** | 0.083 | **Memory retrieval** | 0.297 |
| **Default mode** | 0.067 | Cingulo-opercular | 0.245 |
| **Ventral attention** | 0.064 | Subcortical | 0.232 |
| Visual | 0.054 | **Default mode** | 0.231 |
| Cingulo-opercular | 0.050 | **Auditory** | 0.206 |
| Fronto-parietal | 0.049 | Somatomotor Hand | 0.181 |
| Subcortical | 0.046 | Fronto-parietal | 0.176 |
| Somatomotor Hand | 0.044 | Salience | 0.164 |
| Cerebellar | 0.039 | **Ventral attention** | 0.155 |
| Somatomotor Mouth | 0.036 | Visual | 0.146 |
| **Auditory** | 0.034 | Dorsal attention | 0.141 |
| Dorsal attention | 0.031 | Cerebellar | 0.127 |
| Salience | 0.030 | Somatomotor Mouth | 0.114 |

Table 6: Modules' difference score $T$ of our learnable and Pearson graphs on the ABCD dataset.

| Learnable Graph | | Pearson Graph | |
|---|---|---|---|
| Module | Difference Score | Module | Difference Score |
| **Default mode** | 0.301 | **Default mode** | 0.412 |
| **VentralSalience** | 0.288 | **VentralSalience** | 0.404 |
| CentralExecutive | 0.275 | DorsalSalience | 0.368 |
| DorsalSalience | 0.221 | CentralExecutive | 0.347 |
| **Somatomotor** | 0.217 | Visual | 0.322 |
| Visual | 0.165 | **Somatomotor** | 0.301 |

