# OpenReview forum: "FBNETGEN: Task-aware GNN-based fMRI Analysis via Functional Brain Network Generation"
_MIDL.io/2022/Conference — MIDL 2022_

### Official Review · Reviewer_eiwX · 2022-01-09

**Confidence:** 4
**Preliminary Rating:** 4
**Recommendation:** Poster

**Summary:**

The authors introduce a novel GNN-based framework for functional MRI analysis where functional brain networks were generated by the model and later on used to predict clinical outcomes given fMRI BOLD signal series. The experimental evaluation on different datasets showed the validity of the proposed architechture.

**Strengths:**

As follows:

The paper is well structured and well written.
The description of the proposed method is clear.
Experiments on two real-world datasets showed the validity of the proposed method via extensive comparisons.
Well-developed appendix with the important points missing in the paper (literature and other experimental results).

**Weaknesses:**

As follows:

Explanation of the choice of the GNN needs to be clarified better.
GNN (in general) tends to overfit when learning on brain connectivities. How was this considered when training the framework is a point to be clarified in the paper.
Discussion of the advantage and limitations of the presented framework is missing.
Lack of discussion of the robustness of the framework when trained on different types of brain connectivity (with different numbers of nodes and edges).

**Deanonymize Review:**

no

**Final Rating After The Rebuttal:**

4: Weak Accept

**Justification Of The Final Rating:**

While the authors explained well the points I mentioned in the review, I see that the paper is not worth publishing in MIDL (maybe another conference). It needs more work: better figure presentation, better argumentation in the introduction, more experiments...

**Paper Type:**

methodological development

**Questions To Address In The Rebuttal:**

In the paper, the authors mentioned "The softmax operation highlights the strong ROI connections by generating skewed positive
edge weights, which are compatible with GNNs and valuable for interpretation." which fits the GCN architecture. How can the positive values be interpreted from a neuroscience perspective? Why not simply make the negative values as zero?

There is a wide variety of papers published related to GNN in network neuroscience:
"Bessadok, A., Mahjoub, M. A., & Rekik, I. (2021). Graph Neural Networks in Network Neuroscience. arXiv preprint arXiv:2106.03535.". I suggest adding more papers related to this particular area of application and discussing the advantages/drawbacks of the proposed GNN-based approach and the previous works.

**Special Issue:**

yes

---

### Official Review · Reviewer_rERf · 2022-01-24

**Confidence:** 3
**Preliminary Rating:** 4
**Recommendation:** Poster

**Summary:**

The paper proposes FBNetGen, a deep learning model which learns functional connectivity matrices (FCMs) from BOLD fMRI time series for use in downstream classification tasks in an end-to-end framework. The key components of the model are (1) a feature encoder for extracting features from BOLD fMRI time series (CNN/GRU), (2) a graph generator that builds task-orientated FCMs, and (3) a graph classifier (GNN). The model is evaluated on the task of gender classification using two publicly available datasets: Philadelphia
Neuroimaging Cohort (PNC) and Adolescent Brain Cognitive Development Study (ABCD). The novelty of the model is in the use of intra and inter regularisation losses to encourages the model to learn gender discriminative FCNs.

**Strengths:**

The paper is clearly written and well motivated. A major strength of the paper is that it performs many experiments comparing FBNetGen to existing models, in particular learnable graph baselines. In addition an ablation study is performed to show the effect of the proposed regularisation losses on classification performance. The code is open sourced and the datasets are publicly available.

**Weaknesses:**

The novelty of the paper is limited. Learning the adjacency matrix of a graph from times series data has been done perviously using a very similar formula as outlined in the paper [1]. In addition, the first paper to learn FCMs from BOLD fMRI data is neither used as a deep learning baseline nor discussed in related work [2]. Pearsons correlation is not the only statistical method for computing FCMs. What about partial correlation which seems to be more widely used in fMRI FCM studies?

[1] Zonghan Wu, Shirui Pan, Guodong Long, Jing Jiang, Xiaojun Chang, and Chengqi Zhang. 2020. Connecting the Dots: Multivariate Time Series Forecasting with Graph Neural Networks. Proceedings of the 26th ACM SIGKDD International Conference on Knowledge Discovery & Data Mining. Association for Computing Machinery, New York, NY, USA, 753–763

[2] Riaz A. et al. (2017) FCNet: A Convolutional Neural Network for Calculating Functional Connectivity from Functional MRI. In: Wu G., Laurienti P., Bonilha L., Munsell B. (eds) Connectomics in NeuroImaging. CNI 2017. Lecture Notes in Computer Science, vol 10511. Springer

**Deanonymize Review:**

no

**Detailed Comments:**

- Improve formatting of Table 1 such as the spacing which is inconsistent and add more detail to caption. Is it +/- standard error or standard deviation? How many runs etc.
- In Section 3.3 is the test statistic used for the t-tests the formula found in G.1. It is not clear from Table 4 in Appendix G.2 if the results are showing just the difference score or a statistically significant difference based on a different test statistic?
- Figure 2 there is no colour bar for the heat maps. Furthermore, the FCMs calculated using Pearson's correlation should be between -1 and 1 but it looks like the results have been thresholded to be positive?
- It is good to see the code is open sourced but the experiments cannot be reproduced without at least a sample of the *pre-processed data*.

**Final Rating After The Rebuttal:**

5: Strong Accept

**Justification Of The Final Rating:**

The authors addressed all of my comments satisfactorily. The motivation of the paper is clear, extensive experiments have been performed, and the code is open source. Overall the proposed method is useful for the community and I believe warrants publication at MIDL 2022.

**Paper Type:**

methodological development

**Questions To Address In The Rebuttal:**

- Add FCNet [1] as a baseline as it is the first deep learning paper to learn FCM from BOLD fMRI data for classification and interpretability.

[2] Riaz A. et al. (2017) FCNet: A Convolutional Neural Network for Calculating Functional Connectivity from Functional MRI. In: Wu G., Laurienti P., Bonilha L., Munsell B. (eds) Connectomics in NeuroImaging. CNI 2017. Lecture Notes in Computer Science, vol 10511. Springer

**Special Issue:**

no

---

### Official Review · Reviewer_X4Rj · 2022-01-26

**Confidence:** 3
**Preliminary Rating:** 4
**Recommendation:** Poster

**Summary:**

The authors proposed FBNetGen, an fMRI analysis framework which is aware of downstream tasks and is also able to highlight the brain region related to the prediction. In the FBNetGen framework, a time series data is encoded, and using the time-series features after encoding, a graph generator is learned to produce a task-oriented functional brain network, and a GNN-based graph predictor is used to perform predictions. The method is evaluated on two fMRI datasets for the task of gender prediction, and is compared to existing works to show its advantage over the other methods.

**Strengths:**

- The paper has a clear structure and well-written, it is overall easy to follow.

- The method is described with details, and enables end-to-end training and interpretability of prediction outcomes.

**Weaknesses:**

- The details of GNN are not given, with a different choice of GNN structures, will it affect the prediction? How to determine a suitable structure for GNN?
- The motivation for the graph generator is to avoid negative weights for edges, could the authors give some explanation of the proposed design, such as the softmax of the features?

**Deanonymize Review:**

no

**Detailed Comments:**

Please see the weakness section.

**Final Rating After The Rebuttal:**

5: Strong Accept

**Justification Of The Final Rating:**

I'd like to thank the authors for giving detailed explanation for each concern with a good knowledge of related works. Overall the paper is well-written and the method is supported by sufficient experiments. I would like to suggest a strong accept.

**Paper Type:**

methodological development

**Questions To Address In The Rebuttal:**

The paper is overall well-written, however there are some concerns regarding the details of graph predictor and intuition behind the design of the graph generator.
Additionally, to avoid negative weights in the edges, has there been existing work to deal with it?

**Special Issue:**

yes

---

### Meta-Review · Area_Chair_GyGL · 2022-02-19

**Recommendation:** Accept (Poster)
**Confidence:** 4

**Metareview:**

Reviewers were generally in consensus with all acceptance ratings. While originally reviews had some concerns of lack of novelty and missing details/explanation for some points, author response to these points generally satisified reviewers. Main strengths of the work include 1) extensive experiments, 2) open source code, 3) well written paper.

---

### Decision · Program_Chairs · 2022-02-28

Accept